# Vector-ICL: In-context Learning with Continuous Vector Representations

**Yufan Zhuang**[1], **Chandan Singh**[2], **Liyuan Liu**[2], **Jingbo Shang**[1], and **Jianfeng Gao**[2]

[1]UC San Diego    [2]Microsoft Research

## Abstract

Large language models (LLMs) have shown remarkable in-context learning (ICL) capabilities on textual data. We explore whether these capabilities can be extended to continuous vectors from diverse domains, obtained from black-box pretrained encoders. By aligning input data with an LLM's embedding space through lightweight projectors, we observe that LLMs can effectively process and learn from these projected vectors, which we term Vector-ICL. In particular, we find that pretraining projectors with general language modeling objectives enables Vector-ICL, while task-specific finetuning further enhances performance. In our experiments across various tasks and modalities, including text reconstruction, numerical function regression, text classification, summarization, molecule captioning, time-series classification, graph classification, and fMRI decoding, Vector-ICL often surpasses both few-shot ICL and domain-specific model or tuning. We further conduct analyses and case studies, indicating the potential of LLMs to process vector representations beyond traditional token-based paradigms.

## 1 Introduction

In-context learning (ICL) has emerged as a powerful paradigm in large language models (LLMs), allowing generalization from limited examples within a given context (Brown et al., 2020; OpenAI, 2023). By providing demonstrations in the context during inference, ICL allows models to adapt to new tasks and formats without the need for retraining. However, since LLMs are trained on discrete natural language tokens, ICL is generally learned and used through natural language, limiting its applicability to non-textual data.

We explore whether LLMs can perform ICL directly on continuous vectors, a capability that could dramatically expand their applicability. Many data modalities, such as sensor readings, financial time series, or scientific measurements, lack a natural text representation. Moreover, even for text data, information like numbers might be better represented via continuous vectors than tokens.

In our study, we observe that LLMs can indeed understand and process continuous context via embedding projection. This technique, which we term Vector-ICL, acts as a bridge between continuous data and the LLM's embedding space. Simple linear projections are often sufficient, though for cross-modal tasks—such as those involving non-textual data like time-series or graphs, non-linear transformations may be required. We demonstrate that training the embedding projector using a straightforward next-token prediction objective enables Vector-ICL, effectively teaching the LLM to "read" continuous vectors. Moreover, fine-tuning the projector on downstream tasks further enhances the effectiveness of continuous context, outperforming few-shot ICL and domain-specific models or tuning.

Our investigation begins with the task of text reconstruction, where we assess whether LLMs can recover information encoded in text embedding. This serves as a proof-of-concept for Vector-ICL, showing that LLMs can indeed extract meaningful information from projected continuous vectors. We then turn to the more complex challenge of arithmetics. Although state-of-the-art LLMs can solve Olympiad mathematical problems (Trinh et al., 2024; OpenAI, 2023), they struggle with pre-

Code is available at: `https://github.com/EvanZhuang/vector-icl`.

**(a) Regular few-shot in-context learning**

| Text | AA |
| Numbers | 1 |

> Input: [Text Snippet 1]. The input's sentiment is positive.
> Input: [Text Snippet 2]. The input's sentiment is negative.
> Input: [Text Snippet 3]. The input's sentiment is ......
>
> **Regular ICL**

**(b) Vector in-context learning**

| Text | AA |
| Numbers | 1 |
| Brain fMRI | |
| Time Series | |
| Graphs | |

Input Data (Texts are used as examples here) → Encoder → $f_{enc}(x) \in \mathbb{R}^{d_{enc}}$ → Projector → $\square \in \mathbb{R}^{d_{dec}}$

> $\square_1$'s sentiment is positive. $\square_2$'s sentiment is negative. $\square_3$'s sentiment is .......
>
> **Vector-ICL**

Figure 1: **Comparing regular in-context learning to vector in-context learning.** (a) In regular ICL, textual demonstrations are given as context during LLM inference. (b) In Vector-ICL, the input space is extended across multiple modalities. The input data is first encoded as embeddings, then transformed into continuous vectors which represent as box tokens ($\square$) via embedding projection. During inference, we provide box tokens in prompts as demonstrations for ICL. We consider box tokens representing text, numerical data, brain fMRI, time series, and graphs in this study.

cise number processing due to the limitations inherent in their tokenization schemes. Our results demonstrate that Vector-ICL offers a more effective approach for function approximation, particularly for large numbers that span multiple tokens, potentially opening new avenues for enhancing LLMs' numerical reasoning capabilities.

Finally, we extend our analysis to a broad range of modalities and tasks, including text classification, summarization, molecule captioning, brain fMRI reconstruction and classification, time-series classification, and graph classification. Across these diverse domains, LLMs exhibit competitive and often superior performance when employing Vector-ICL, revealing previously untapped capabilities of these models. This work highlights the potential of continuous representations in enhancing LLMs' in-context learning capacities, pushing the boundaries of what these models can achieve beyond token-based paradigms.

## 2 RELATED WORK

**Empirical results of in-context learning** ICL has empirically shown strong performance in diverse natural-language processing tasks with very few demonstrations (Brown et al., 2020; OpenAI, 2023). In modern LLMs with long context windows, ICL has even shown performance improvements as the number of demonstrations grows to hundreds or even thousands, sometimes outperforming finetuning (Agarwal et al., 2024; Li et al., 2023a; Bertsch et al., 2024). Empirically, different factors play key roles in ICL. In smaller LLMs, ground-truth demonstrations are not required for in-context learning, while other factors such as the label space, input text distribution, and overall sequence format play an important role (Min et al., 2022b). Moreover, these LLMs can sometimes achieve strong performance even when demonstrations are intentionally irrelevant or even pathologically misleading (Webson & Pavlick, 2022). Flan-T5 (Chung et al., 2024) revealed that instruction tuning improves few-shot learning by helping LLMs better utilize in-context examples in an encoder-decoder architecture.More recently, Wei et al. (2023) characterize these behaviors of LLMs with respect to model size, and show that larger language models perform in-context learning differently in the presence of flipped or semantically unrelated labels. Orthogonally, different works find ways to improve ICL, e.g. by including explanations (Lampinen et al., 2022), or chaining ICL calls (Morris et al., 2023). ICL has shown some success in multimodal models (Wu et al., 2024; Jiang et al., 2024) or when applied to tabular data (Zhao et al., 2024).

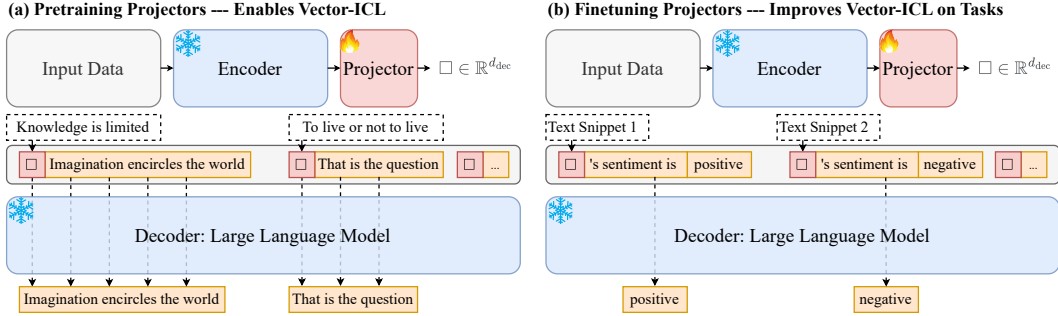

Figure 2: **Pretraining and finetuning the projectors.** Vector-ICL requires updating the parameters of a lightweight projector while keeping the encoder and decoder parameters fixed. The encoder first compresses the input into single token embeddings, and then the projector will project it to the aligned representation space for LLMs' later use. (a) Pretraining the projector on a general language modeling corpus (or a modality-to-text dataset) enables Vector-ICL. (b) Task-specific fine-tuning makes Vector-ICL outperform few-shot ICL on natural language tasks, as well as with domain-specific models on non-language tasks.

**Understanding ICL**  Many works have investigated ICL and found that it is able to learn linear models (Akyürek et al., 2022; Zhang et al., 2023), discrete functions (Bhattamishra et al., 2023), and more general algorithms (Li et al., 2023b). Some works have explicitly connected ICL in specific settings to implementing optimization steps analogous to gradient descent (Mahankali et al., 2023; Von Oswald et al., 2023; Ahn et al., 2024) and higher-order optimization methods (Dai et al., 2023; Fu et al., 2023; Giannou et al., 2024; Zhang et al., 2023). A complementary direction aims to establish statistical complexity and generalization bounds of in-context learning in transformers (Bai et al., 2024; Li et al., 2023b; Wies et al., 2024; Wu et al., 2023). Finally, one recent work suggests that ICL may arise from parallel structures in pretraining data (Chen et al., 2024).

**Learning to learn in-context**  In contrast to the emergent ICL capabilities of LLMs, existing works have also studied how to explicitly improve ICL. Min et al. (2022a) propose MetaICL, a meta-training framework for finetuning pretrained LLMs to perform in-context learning on a large and diverse collection of tasks. In the tabular domain, TNP (Nguyen & Grover, 2022) and PFNs (Müller et al., 2021) train transformer models to perform in-context prediction for a family of functions, which allows in-context generalization to unseen functions after training. Zhao et al. (2023) also propose meta-learning transformers to in-context learn group preferences, serving as an in-context learned reward model that adapts to diverse group preferences.

## 3  METHOD: VECTOR CONTEXT VIA EMBEDDING PROJECTION

### 3.1  EMBEDDING PROJECTION

Vector-ICL requires transforming inputs into vector contexts through an embedding projection. Given a dataset $\mathcal{X} = \{x_i\}_{i=1}^n$, we assume the existence of an encoder $f_{\text{enc}}$, that transforms the data into an abstract representation (alternatively, the raw data may already be a continuous vector). The encoded embeddings, $f_{\text{enc}}(x)$, are then projected into box tokens, denoted as $\square_x$. Throughout the paper, we will use the terms "box tokens" and "projected embeddings" interchangeably. For decoding, we use a language model $f_{\text{llm}}$ that generates outputs based on the provided prompts.

We impose no constraints on the form of the input data $\mathcal{X}$; it can come from any modality. The only requirement is that the encoder $f_{\text{enc}}$ maps each data point $x$ into a vector space, defined as:

$$f_{\text{enc}} : x \to \mathbb{R}^{d_{\text{enc}}}, \ \forall x \in \mathcal{X} \tag{1}$$

The LLM typically processes discrete tokens $\{\text{tok}_1, \text{tok}_2, \dots, \text{tok}_l\}$, then maps them to text embedding space $\{\text{emb}_1, \text{emb}_2, \dots, \text{emb}_l\}$, $\forall i, \ \text{emb}_i \in \mathbb{R}^{d_{\text{dec}}}$. Since we operate mostly in embedding space, we omit the tokenization step for simplicity and directly refer to text inputs as their embedding representations.

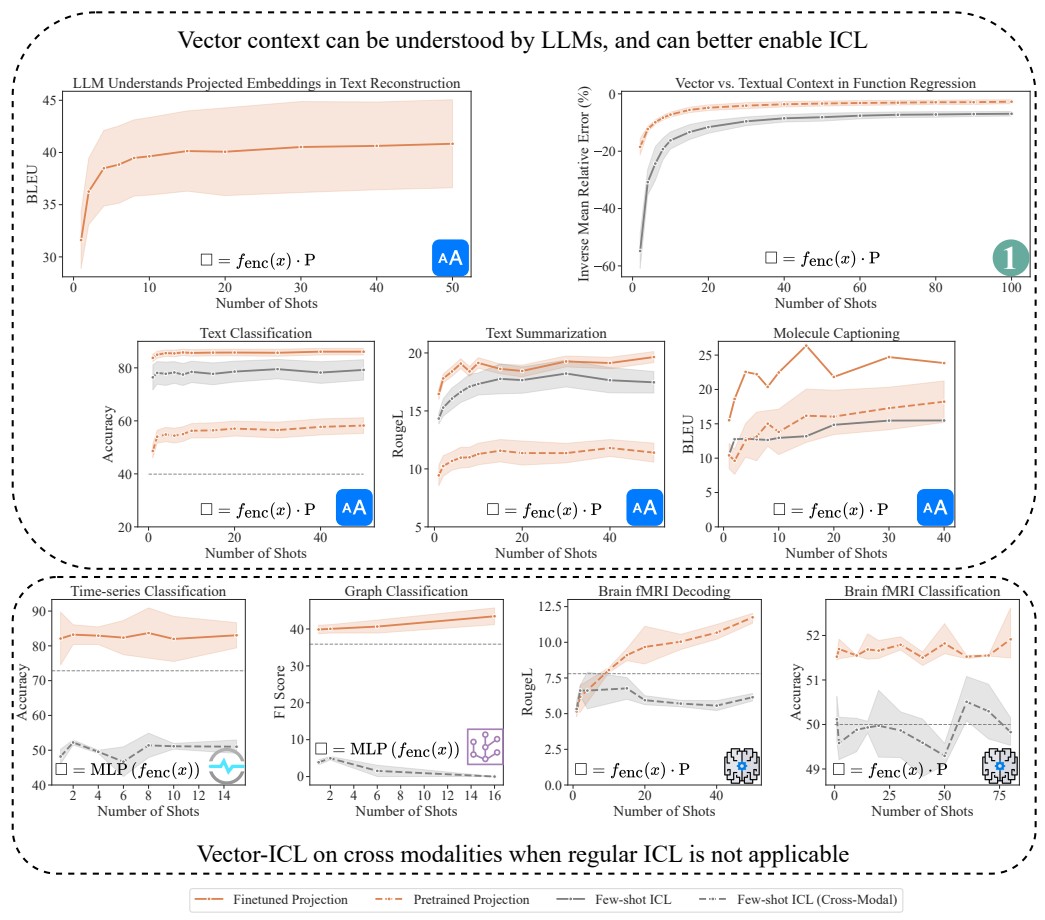

Figure 3: **Main results: LLMs can perform Vector-ICL ($\uparrow$ = better).** We show that training the embedding projector with a simple next-token prediction objective enables Vector-ICL. Even with only unsupervised pretraining, Vector-ICL matches or outperforms traditional few-shot ICL on 4 out of 6 tasks where direct comparison is possible. Fine-tuning the projector on downstream tasks further enhances the use of continuous context, consistently surpassing both few-shot ICL and specialized task-tuned baselines (soft-prompt for text, tuned encoders for non-text).The study begins with text reconstruction to assess LLMs' ability to interpret box token embeddings, followed by function regression to evaluate reasoning capabilities. We then demonstrate Vector-ICL's effectiveness and applicability across various downstream tasks, including text classification, summarization, time-series classification, graph classification, and brain fMRI decoding & classification. Results in each panel are averaged over different encoders and LLMs for the diverse tasks we study; error bars show 95% confidence intervals.

The process of embedding projection is then carried out as follows. For linear projection, we construct a projection matrix $P \in \mathbb{R}^{d_{\text{enc}} \times d_{\text{dec}}}$ and make the following transformations to obtain projected embedding $\square_x$ given input $x$:

$$\square_x := f_{\text{enc}}(x) \cdot P \tag{2}$$

In cases where more expressive power is needed, we utilize a two-layer multi-layer perceptron (MLP) to perform the projection:

$$\square_x := \text{MLP}(f_{\text{enc}}(x)) \tag{3}$$

The MLP follows the architecture of the MLP block in Llama (Touvron et al., 2023), with an additional input projection layer to map the input dimension $\mathbb{R}^{d_{\text{enc}}}$ to output dimension $\mathbb{R}^{d_{\text{dec}}}$.

## 3.2 PROJECTED EMBEDDINGS AS CONTEXT

The projected embeddings are then utilized as context in Vector-ICL, functioning as the equivalent of the original input data. For example, in NLP tasks, the original text snippets $x$ will first be encoded as embeddings $f_{\text{enc}}(x)$, then projected to become $\square_x$, inserted into the prompt like the following:

$$\square_x\text{'s sentiment is } \ldots / \square_x\text{'s summarization is } \ldots$$

Using them as the context in ICL is then natural:

$$\square_1\text{'s sentiment is happy.} \quad \square_2\text{'s sentiment is sad.} \quad \square_x\text{'s sentiment is } \ldots$$

where $\square_1$ and $\square_2$ are in-context examples and $\square_x$ is the input.

## 3.3 TRAINING THE EMBEDDING PROJECTORS

The projectors need to be trained to achieve effective projections. We discovered that pretraining these projectors with language modeling objectives enables the ICL capabilities with vector context, and finetuning them on task datasets further improves ICL performance.

The pretraining process is depicted in Fig. 2(a). For each text snippet, we cut it into two pieces with the cutting point randomly sampled from the end of sentences. The first half is encoded and projected while the second is kept intact. The rest is the same with any pretraining process, the language model generated the next token distribution at each input position, except for the ones preceding the projected embeddings, and a cross-entropy loss is imposed on top of this. With the encoder and LLM frozen, the gradient backpropagates to the projector, updating its parameters.

For non-text data modalities, pretraining can be more flexible. We define this pretraining as involving general, non-task-specific objectives, such as reconstructing a number from its embeddings (e.g., $\square_x$ is $\underline{32768}$), performing basic algebra (e.g., $\square_x + \square_y = \underline{16384}$), or predicting the next token from brain fMRI embeddings.

The finetuning process is shown in Fig. 2(b). It utilizes additional structured prompts and trains with task-specific datasets. Similarly, the input is first mapped into the embedding space and projected into $\square$ tokens. They are then inserted into structured prompts, while the projector is trained with conditional generation loss given those prompts.

## 4 EXPERIMENTAL SETUP

Table 1 gives an overview of our experimental setup, including specifics for the task, datasets, encoders, LLMs, and task-specific prompts we use. Across different tasks, we project to four open-weights LLMs. We now provide details for individual tasks.

**Baselines**  We evaluate Vector-ICL in two distinct settings: with and without task-specific tuning. For textual tasks, we compare against few-shot ICL and soft prompt tuning (Li & Liang, 2021). We choose soft prompt tuning as our primary baseline because it represents a similarly lightweight adaptation approach - both methods introduce a small number of trainable parameters while keeping the base LLM frozen. Like our projector, soft prompts modify how the LLM processes inputs without changing its internal weights. This makes it a fair comparison point for assessing whether Vector-ICL's benefits come from the continuous vector representations themselves rather than just the additional training. For non-textual tasks, where soft prompts cannot be directly applied, we compare against cross-modal few-shot ICL and tuned encoders. In the non-textual domain, ICL inputs are represented either as numeric sequences (for time-series and brain fMRI data) or as textual descriptions (for graph edge lists and node features).

**Text Pretraining**  To pretrain our text projectors, we leverage the WikiText-103 (Merity et al., 2016) dataset, consisting of over 100 million tokens from verified high-quality Wikipedia articles. This smaller language modeling corpus is chosen for its suitability to the lightweight nature of our projectors. The pretraining process is illustrated in Fig. 2(a), where text snippets are divided at random sentence-end points. The first half is embedded and projected, while a next-token generation loss is applied to the second half.

Table 1: **Overview of the tasks, datasets, encoders, large language models, and prompts used in the experiments.** Each task utilizes encoders and LLMs to perform functions across multiple modalities. The table highlights the diversity of models and configurations applied to each task.

| Task | Dataset | Encoder | LLM | Prompt |
|---|---|---|---|---|
| Text Reconstruction | Quora (Thakur et al., 2021) PST (Tiedemann, 2012) | N, S, S, G | 🦙 | Translate the text in brackets: (□) Translation: [Text] |
| Function Regression | 10 Digits Regression | Digit Embedding | 🦙, Ⓜ, 🐤, Ⓨ | $x = □_x, \ y = □_y,$ function$(x, y)$ equals to (digits): [Solution] |
| Text Classification | IMDB (Maas et al., 2011) Rotten Tomatoes (Pang & Lee, 2005) SST2 (Socher et al., 2013) Emotion (Saravia et al., 2018) Financial Phrasebank (Malo et al., 2014) | N, S, S, G | 🦙, Ⓜ, 🐤 | (□)'s sentiment is: [Label] |
| Text Summarization | XSum (Narayan et al., 2018) XLSum (Hasan et al., 2021) | N, S, G | 🦙 | (□)'s summarization is: [Summary] |
| Molecule Captioning | Language + Molecule-24 (Edwards et al., 2024) | N | 🦙 | (□)'s molecule caption is: [Caption] |
| Brain fMRI | LeBel et al. 2022, Tang et al. 2023 | PCA of fMRI | 🦙 | [Question] Input: □ Response: [Answer] |
| Time-series Classification | FordA, FordB (Dau et al., 2019) | Chronos-base (Ansari et al., 2024) | 🦙 | (□)'s class (positive, negative) is: [Label] |
| Graph Classification | ogbg-molhiv (Hu et al., 2020) | Graphormer (Ying et al., 2021) | 🦙, 🐤 | (□)'s class (positive, negative) is: [Label] |

**Encoders:** N NV-Embed (Lee et al. 2024; `nvidia/NV-Embed-v1`), S SFR (Meng et al. 2024; `Salesforce/SFR-Embedding-2_R`), S Stella (Zhang 2024; `dunzhang/stella_en_1.5B_v5`), G GTR-t5 (Ni et al. 2021; `sentence-transformers/gtr-t5-base`)
**LLMs:** 🦙 Llama-3.1-8B (Dubey et al. 2024; `meta-llama/Llama-3.1-8B-Instruct`), Ⓜ Mistral-7B (Jiang et al. 2023; `mistralai/Mistral-7B-Instruct-v0.3`), 🐤 Qwen2-7B (Yang et al. 2024; `Qwen/Qwen2-7B-Instruct`), Ⓨ Yi-1.5-9B (Young et al. 2024; `01-ai/Yi-1.5-9B-Chat`)

**Text Reconstruction**   We investigate LLMs' ability to decode original text from projected embeddings using two datasets: Parallel Sentence Talks (Tiedemann, 2012) and Quora (Thakur et al., 2021). These datasets consist of concise text pieces that convey clear meaning. Projectors are trained on the training sets of both datasets, and performance is evaluated using the BLEU score (Papineni et al., 2002; Post, 2018).

**Arithmetic and Function Regression**   For the arithmetic tasks, we generated synthetic datasets containing 10-digit numbers, which are particularly challenging for LLMs as they require splitting the numbers into multiple text tokens. These numbers are represented using a concatenated one-hot encoding per digit. For instance, a 10-digit number is represented as a $10 \times 10$ matrix, flattened into a 100-dimensional vector. The pretraining phase includes two key tasks: number reconstruction, where the model is tasked with recovering the original number from its embedding, and basic arithmetic, where the model performs algebraic addition operations on the projected embeddings.

To evaluate the models' arithmetic reasoning abilities, we employ a non-linear function regression task, where the function is defined as $f(x, y) = \sqrt{x}\sqrt{y}$. The model is provided with inputs $x$ and $y$, and it must predict the integer part of the function output. Performance is measured using the mean relative error, calculated as the $\ell_1$ difference between the predicted and true values, normalized by the ground truth. This task allows us to assess the models' ability to perform more complex numerical reasoning beyond simple arithmetic operations.

**Text Classification**   We assess whether Vector-ICL can be applied effectively to text classification. Both binary and multi-class classification datasets are used, and the results are compared across few-shot ICL and soft prompt tuning. The classification performance is measured by accuracy.

**Text Summarization**   Following the classification tasks, we explore Vector-ICL's capability in summarizing text based on the projected embeddings. The datasets, encoders, LLMs, and prompt templates can be found in Table 1. Performance is evaluated using RougeL (Lin, 2004).

**Molecule Captioning**   We also extend our approach to the unconventional task of molecule captioning, using molecule sequence-caption pairs from the Language + Molecules-24 (LPM24) (Edwards et al., 2024) dataset. A sample molecule-caption pair looks like the following:

Table 2: Comparison of Vector-ICL, few-shot ICL, and soft prompt tuning across various sentiment analysis and summarization datasets. Details can be found in Appendix A.3.

| Method | Sentiment Analysis | | | | | Summarization | |
|---|---|---|---|---|---|---|---|
| | Rotten Tomatoes | SST2 | IMDB | Emotion | Financial Phrasebank | XSum | XLSum |
| V-ICL (pretrained) | 80.60 | 78.90 | 95.04 | 41.20 | 60.72 | 15.25 | 15.89 |
| Few-shot ICL | 87.31 | 91.74 | 93.50 | 55.20 | 71.78 | 19.53 | 19.41 |
| Soft Prompt | **93.24** | 96.21 | 95.26 | 74.15 | 78.22 | 12.84 | 12.70 |
| V-ICL (finetuned) | 88.80 | **98.16** | **97.28** | **85.20** | **81.68** | **20.08** | **20.49** |

Molecule: Cc1c(Cl)cccc1-n1ccn2c(SCC(=O)c3ccccc3C(F)(F)F)nnc2c1=O
Caption: The molecule is a pain treatment that impacts inflammatory disease treatment.

This task explores whether LLMs can extract useful information from projected embeddings of out-of-distribution chemical sequences, with performance evaluated via BLEU score.

**Brain fMRI Decoding and Classification** We analyze data from LeBel et al. 2022 and Tang et al. 2023, which consists of fMRI responses for 3 human subjects as they listen to 20+ hours of narrative stories from podcasts. We preprocessed the data following Benara et al. 2024, by converting the fMRI responses into a 200-dimensional output using principal components analysis and assigning classification labels to 10-grams of the story text at 2-second intervals using an LLM.

We use the same pretraining methodology as text to pretrain on the brain fMRI data (projecting on 20% of time points and imposing next-token generation loss on the remaining 80%) . We evaluate the LLM's capability to decode projected brain fMRI by giving them randomly sampled context from the train set, that could come from different human subjects or from a different story, and ask them to decode segments from the test set.

In addition to text reconstruction, we decode the binary labels from the fMRI responses, corresponding to questions about the underlying text, e.g. "Does the sentence contain a proper noun?" The decoding random baseline is constructed by giving the LLM the randomly sampled, shuffled text from the training set, and generating text according to it. We measure the performance using the RougeL score between the generated text and the ground truth text. The classification random baseline is 50% accuracy, as we have balanced the dataset.

**Time-series** We take the output of the last time step from Chronos-base (Ansari et al., 2024) as the time-series' representation. We use the base encoder with trained classification head as the baseline and we measure the prediction performance with accuracy.

**Graphs** We use Graphormer (Ying et al., 2021) as the encoder model, specifically the one that was pretrained on quantum chemistry graph datasets (Hu et al., 2021). Since the down-stream, ogbg-molhiv (Hu et al., 2020), is a molecule property prediction dataset, and with strong class imbalance (3-4% positive classes), we finetune the encoder on the training set to provide meaningful baselines and embeddings. We take the output prior to the classification layer of the Graphormer as the graph embedding. Weighted sampling is adopted in the finetuning of both the baseline Graphormer and the embedding projector to yield meaningful predictions. We use the finetuned Graphormer as our baseline and use the F1 score as the performance metric.

**Projector Configurations** Both linear and non-linear projectors are utilized, as shown in Fig. 3, with input and output dimensions matching the encoder-decoder pairs. Early stopping with patience of 500 steps is used during finetuning, as projectors converge quickly due to their small parameter sizes. Details of the hyperparameters used in training are provided in Table 3.

## 5 RESULTS: UNLOCKING VERSATILE APPLICATIONS ACROSS MODALITIES

Fig. 3 presents our main results, where each subplot corresponds to one of the nine tasks. We begin by exploring text reconstruction, to see whether LLMs can comprehend the information encoded

within the box tokens. Next, we investigate the tasks of function regression to evaluate whether LLMs can leverage the box tokens during reasoning processes and whether this approach outperforms reasoning with plain text. Finally, we proceed to a range of downstream tasks, including text classification, text summarization, time-series classification, graph classification, and brain fMRI decoding. This comprehensive evaluation allows us to assess the versatility and effectiveness of Vector-ICL across different domains and task types.

**Text Reconstruction: LLM Understanding of Projected Embeddings**  We first verify LLMs' ability to understand projected embeddings. Our results demonstrate that Vector-ICL successfully reconstructs original text from projected embeddings, with performance improving as the number of examples (shots) increases, mirroring standard in-context learning (ICL) behavior. This suggests that LLMs can effectively decode the information compressed into the box tokens, with more context leading to better reconstruction. The similarity to ICL behavior indicates that Vector-ICL leverages similar learning mechanisms, but with the flexibility of working with continuous representations.

**Function Regression: Enhanced Reasoning with Continuous Context**  We then study would it sometimes be better for LLMs to receive continuous context instead of discrete tokens. After pretraining projectors on 10-digit number reconstruction and addition between two numbers, we task the LLM with learning an unknown function in-context. Results show Vector-ICL consistently outperforms few-shot ICL with raw number inputs, that have to span multiple tokens. This suggests that the continuous representations capture numerical relationships more effectively than discrete tokens, enabling LLMs to better infer and apply mathematical patterns. The improvement is particularly noteworthy given that LLMs are typically challenged by precise numerical computations.

**Text Classification**  In this classical NLP task, we aggregate mean accuracy across five datasets, four encoders, and three LLMs. Results indicate that pretrained projectors provide meaningful continuous context for ICL, outperforming the random baseline. LLMs achieve optimal performance with continuous context from finetuned projectors, surpassing both regular few-shot ICL and soft prompt tuning, with details shown in Table 2. This demonstrates the versatility of Vector-ICL across different text classification scenarios and its ability to outperform established prompt tuning methods. The success across multiple datasets and LLMs suggests that the benefits of continuous context are robust and generalizable.

**Text Summarization**  This task demands longer text generation and deeper information comprehension. We aggregate mean RougeL scores across two datasets, and three encoders. Findings show that pretrained projectors enable LLMs to extract and condense information from a single box token, while finetuned projectors provide more effective continuous context than original textual input and soft prompts, with details shown in Table 2. The ability to compress and later expand information from a single token is particularly impressive, suggesting that Vector-ICL captures high-level semantic content effectively. The superior performance of finetuned projectors highlights the benefits of task-specific optimization in continuous space.

**Molecule Captioning: An Unconventional NLP Task**  We explore LLMs' ability to comprehend continuous vector context for out-of-distribution inputs like molecule sequences. Evaluating captioning performance with BLEU scores, we find that both pretrained and finetuned projectors provide better context than original molecule sequence text, despite encoders likely never trained on such sequences. This result is particularly intriguing as it demonstrates Vector-ICL's ability to bridge the gap between specialized domains and general language understanding. It suggests that continuous representations can capture and translate domain-specific information in a way that's more accessible to LLMs than raw specialized notation.

**Time-series Classification**  We finetune non-linear projectors on the training sets of two datasets and evaluate LLM performance with the resulting continuous context. Aggregating average accuracy, we find LLMs outperform baseline domain-specific models with finetuned classification heads. The success here suggests that continuous context can effectively capture temporal dependencies and patterns, translating time-series data into a form that LLMs can process effectively.

**Graph Classification**    Since the task dataset is heavily imbalanced and out of the pretraining distribution of the graph encoder. We first finetune the base encoder on the target dataset to establish a baseline, then train non-linear projectors on the graph classification dataset. Averaging F1 scores across two LLMs, results indicate that Vector-ICL enables LLMs to outperform the finetuned baseline model. This is a noteworthy achievement, as graph data is structurally very different from the text data that LLMs are trained on. The success here suggests that Vector-ICL can effectively translate graph structures into continuous representations that preserve relational information in a way that's interpretable to LLMs.

**Brain fMRI Decoding and Classification**    We pretrain projectors on the training set using next-token generation loss, then apply them to recover original text from brain fMRI signals in the test set. Results show LLMs can surpass random baselines by leveraging projected 200-dimensional fMRI PCA factors, with performance improving as context increases. This application to neuroscience data is particularly exciting, demonstrating Vector-ICL's potential in bridging neural activity and language understanding.

## 6    ANALYSIS

### 6.1    THE IMPACT OF ENCODER QUALITY ON VECTOR-ICL PERFORMANCE

We investigate the relationship between the intrinsic capabilities of encoders and their effectiveness when used with Vector-ICL for downstream tasks. To quantify the encoders' intrinsic abilities, we evaluate their performance on a text reconstruction task, which serves as a proxy for the amount of information preserved in the embeddings.

Our analysis focuses on text classification as the downstream task. We examine the correlation between encoder rankings on the reconstruction task and their corresponding rankings on the classification task. This analysis is performed across 5 datasets and 3 LLMs, resulting in 15 configurations.

The results of this analysis, presented in Fig. 4a, demonstrate a consistent positive correlation between an encoder's text reconstruction performance and its effectiveness in downstream classification tasks when used with Vector-ICL. Notably, in 4 of the 15 configurations, we observe a particularly strong correlation, with values approaching 1.

Our findings suggest that an encoder's performance on the text reconstruction task can serve as a reliable predictor of its potential effectiveness in downstream tasks when integrated with Vector-ICL. This insight could prove valuable for practitioners in selecting encoders for Vector-ICL.

### 6.2    CASE STUDY: WHAT HAS BEEN LEARNED IN THE PROJECTIONS?

Fig. 4b provides a visualization of the normalized Euclidean distances between projected embeddings. Several key patterns emerge from this that offer insights into what the projector has learned.

Analysis of the numerical embedding distance matrix reveals key properties of our projection method. Embeddings for similar numbers cluster along the diagonal, indicated by lighter colors, demonstrating the preservation of local structure. Conversely, increasing distances from the diagonal, shown by darker colors, indicate effective separation of numerically distant values in the embedding space.

Another notable feature of the distance matrix is the block structure that emerges. The block structure reflects how numbers share similar digit patterns across the decimal places, from local to global blocks. It likely helps LLMs process numerical relationships more effectively, as it preserves the hierarchical nature of place-value notation.

### 6.3    CASE STUDY: CAN SYNTHETIC DATASET BE USEFUL IN CROSS-MODAL PRETRAINING?

The cross-modal pretraining requires data-text dual pairs, which is not available in many cases. We explore how to create a synthetic dataset for such pretraining using time-series as an example. We designed a data curation pipeline that extracts meaningful statistical properties from time-series and converts them into natural language descriptions.

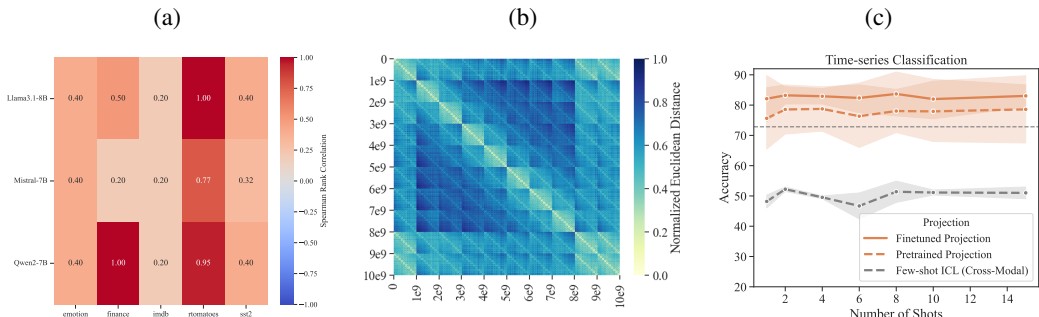

Figure 4: **Key Insights from Encoders, Projections, and Synthetic Data Curation.** (a) Correlation between encoders' text reconstruction performance and their downstream task effectiveness with Vector-ICL, suggesting information preservation ability predicts Vector-ICL performance. (b) Euclidean distance matrix of 1024 projected number embeddings (0 to 1e10) shows structured block-diagonal patterns, indicating meaningful numerical relationships are preserved. (c) Results from pretraining on our synthetic time-series QA dataset, which captures statistical properties through trend analysis, anomaly detection, and stability assessment. The curated QA pairs enable effective cross-modal pertaining.

Our pipeline analyzes multiple statistical aspects of the time-series data. We perform trend analysis through linear regression to capture overall directional movements, anomaly detection using z-score analysis to identify unusual patterns, and temporal stability assessment via variability thresholds to characterize consistency.

For each identified property, we generate both descriptive statements and binary questions. For instance, given a time-series segment, we generate descriptive text like "The time-series shows an upward trend" or binary questions such as "Does this time-series contain any anomalies?"

Our experiments show that projectors pretrained on this synthetic corpus can effectively generalize to new tasks, demonstrating the feasibility of creating synthetic data for cross-modal pretraining when natural data pairs are unavailable.

# 7 DISCUSSION

**Limitations and Future Directions** In this study, we explored a variety of settings: utilizing different encoders, LLM architectures, modalities, and datasets. Our results demonstrate that LLMs are capable of performing Vector-ICL on both language and non-language inputs. However, our experiments did not cover all possible combinations of these variables. There are still many unexplored areas, such as additional modalities, tasks, and encoder-decoder configurations, that could further benefit from Vector-ICL. Also, we only experimented with single-token encoders, while there exist encoders that produce variable-sized embeddings, that can potentially be more powerful and flexible. Analyzing how instruction tuning might affect the model's ability to understand vector context would be beneficial as well. We leave this extensive exploration for future research to fully understand the broader applicability and limitations of our approach across diverse domains.

**Conclusion** In this work, we explore whether large language models trained only on text can perform in-context learning on continuous vectors from different domains. Our findings suggest that LLMs can indeed understand and process continuous context via embedding projection. Simple linear projections are often sufficient, though for cross-modal tasks—such as those involving non-textual data like time-series or graphs—non-linear transformations may be required. In our experiments across various tasks and modalities, including text reconstruction, numerical function regression, text classification, summarization, molecule captioning, time-series classification, graph classification, and fMRI decoding, Vector-ICL often surpasses both few-shot ICL and task-specific model or tuning. We further conduct analyses and case studies, indicating the potential of LLMs to process vector representations beyond traditional token-based paradigms.

ACKNOWLEDGEMENT

Our work is sponsored in part by NSF CAREER Award 2239440, NSF Proto-OKN Award 2333790, as well as generous gifts from Google, Adobe, and Teradata. Any opinions, findings, and conclusions or recommendations expressed herein are those of the authors and should not be interpreted as necessarily representing the views, either expressed or implied, of the U.S. Government. The U.S. Government is authorized to reproduce and distribute reprints for government purposes not withstanding any copyright annotation hereon.

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

# A  APPENDIX

## A.1  DETAILED EXPERIMENT SETUP

**Text Reconstruction**   We use two datasets for the text reconstruction task, Parallel Sentence Talks's English subset (Tiedemann, 2012) and Quora (Thakur et al., 2021).

Parallel Sentence Talks consist of sentences used in movie conversations, and Quora is built on a wide range of online questions. They represent short pieces of text that convey clear information. We aim to explore whether LLMs can decode the original message from the projected text embeddings.

We use the following prompt template:

Translate the text in brackets: ($\square$), translation: [Original Text]

We train the projectors on the training set of these two datasets and evaluate their performance on the test tests. We measure the reconstruction performance with the BLEU score (Papineni et al., 2002; Post, 2018).

**Arithmetic and Function Regression**   We created synthetic datasets of numerical data to pretrain our linear number projectors, experimenting with two configurations: one using 3-digit numbers and the other using 10-digit numbers. In Llama3.1-8B's tokenizer, 3-digit numbers are represented as single tokens, while in Mistral-7B, Qwen2-7B, and Yi-1.5-9B, numbers larger than 10 are split into multiple tokens. Consequently, 10-digit numbers consistently span multiple tokens across all models, which increases the complexity of performing arithmetic operations.

To represent the numbers, we use a concatenated-and-flattened one-hot vector encoding for each digit. For instance, a 3-digit number is represented as a $3 \times 10$ matrix (one hot per digit place), which is then flattened into a 30-dimensional vector. Similarly, a 10-digit number is represented as a $10 \times 10$ matrix, flattened into a 100-dimensional vector.

The pretraining involves two tasks. The first task is number reconstruction, we use the following prompt template, given the number is 128:

$$x = \square_x, \quad x \text{ equals to (digits): } \underline{128}$$

The second task is basic addition, we use the following prompt template, given the numbers are $x = 128 \ y = 256, \ a = 1, \ b = 1$:

$$x = \square_x, \ y = \square_y, \quad a * x + b * y \text{ equals to (digits): } \underline{384}$$

Here, $a$ and $b$ are randomly sampled from $[0, 1]$ with up to two decimal places, and the solution is the integer part of the sum.

For evaluation, we use a function regression task with a non-linear function: $f(x, y) = \sqrt{x} \cdot \sqrt{y}$. The LLM is given inputs $x$ and $y$, along with the integer part of the output $f(x, y)$. The model is then tasked with predicting the output for new pairs of $x$ and $y$. The prompt for in-context learning is structured as follows:

$$x = \square_x, \ y = \square_y, \quad \text{function}(x, y) \text{ equals to (digits): } \underline{[\text{Solution}]}$$

For few-shot ICL, the box tokens will be replaced with the actual numbers. We measure the function regression performance with the mean relative error, where the relative error is computed as the $\ell_1$ difference divided by the ground truth value.

**Text Classification**   We use five datasets for the text classification task. For binary classification, we include IMDB (Maas et al., 2011), Rotten Tomatoes (Pang & Lee, 2005), and the Stanford Sentiment Treebank (SST2) (Socher et al., 2013). For multi-class classification, we use the Emotion dataset (Saravia et al., 2018) and the Financial Phrasebank (Malo et al., 2014). The binary classification datasets (IMDB, Rotten Tomatoes, and SST2) involve classifying movie reviews as positive or negative. The Emotion dataset classifies Twitter tweets into six categories: anger, fear, joy, love, sadness, and surprise. The Financial Phrasebank categorizes financial news into positive, negative, or neutral sentiments.

We use the following prompt in ICL:

$$(\square)\text{'s sentiment is } \underline{\text{[Input Class]}}$$

For few-shot ICL, the box tokens will be replaced with the actual text. For soft prompt tuning, we use 20 virtual tokens and train for one epoch over the entire training set. We measure the classification performance with accuracy on the test set.

**Text Summarization**   We use two datasets for the summarization task, XSum (Narayan et al., 2018) and the English subset of XLSum (Hasan et al., 2021). They contain newspaper articles and their summaries.

We use the following prompt in ICL:

$$(\square)\text{'s summarization is: } \underline{\text{[Summary of the Input]}}$$

For few-shot ICL, the box token will be replaced by the article. We measure the performance using the RougeL score with the ground truth summary on the test sets.

**Molecule Captioning**   We use the Language + Molecules-24 (LPM24) dataset for the molecule captioning task, it was created for the task of molecule-language translation, and contains 161K pairs of molecule strings and their captions in the combined training and test set.

A sample molecule-caption pair looks like the following:

Molecule: Cc1c(Cl)cccc1-n1ccn2c(SCC(=O)c3ccccc3C(F)(F)F)nnc2c1=O
Caption: The molecule is a pain treatment that impacts inflammatory disease treatment.

And we use the following prompt for ICL:

$$(\square)\text{'s molecule caption is: } \underline{\text{[Caption of the Input Molecule]}}$$

For few-shot ICL, we replace the box token with the actual molecule string. We measure the performance using the BLEU score between the generated caption and the ground truth caption.

**Brain fMRI Decoding and Classification**   We analyze data from LeBel et al. 2022 and Tang et al. 2023, which consists of fMRI responses for 3 human subjects as they listen to 20+ hours of narrative stories from podcasts. We preprocessed the data following Benara et al. 2024, by converting the fMRI responses into a 200-dimensional output using principal components analysis and labeling 10-grams of the story text at 2-second intervals using an ensemble of LLMs.

The data was separated into train set and test set by holding out the same three podcast stories from the three human subjects. We use the same pretraining methodology as text to pretrain on the brain fMRI data. As the data comes in as segments of text and the recorded fMRI, we randomly sample 20% of the segments to be in fMRI form and projected into box tokens, and we impose next token generation loss on the rest 80%.

We evaluate the projectors by giving them randomly sampled context from the train set, that could come from different human subjects or from a different story, and ask them to decode segments from the test set. We use the following prompt in ICL in our decoding experiments:

What is the English translation of the input?
Input: $\square$, Response: the input in English is $\underline{\text{[Text Corresponding to fMRI]}}$

The random baseline is constructed by giving LLM the randomly sampled, shuffled text from the training set, and generating text according to it. We measure the performance using the RougeL score between the generated text and the ground truth.

We construct the classification questions around the properties of the underlying text, for example, "Does the sentence contain a proper noun?", ""Does the input mention anything related to arguing?". The ground truth is obtained via GPT4o (OpenAI, 2023) as binary labels. We use the following prompt in ICL in our classification experiments, using one of the example questions:

Does the input mention anything related to arguing?
Input: $\square$, Response (Yes or No): according to the English text of the input,
the answer is $\underline{\text{[Yes/No]}}$

The random baseline is 50%, as we have downsampled and balanced the data. And we use accuracy as the performance metric.

**Time-series**  We use the Chronos (Ansari et al., 2024) time-series Transformers as the encoder. Chronos was pretrained on large scale time-series and is designed to generate the next segments of the time-series. It has proven effective on a wide range of time-series forecasting benchmarks. We take the output of the last time step from Chronos-base as the time-series representation.

We use two datasets for the time-series classification task, FordA, and FordB, they are also part of the UCR Time Series Classification Archive (Dau et al., 2019) ranging from 4000 to 5000 time-series for each dataset. We use the following prompt in ICL:

(□)'s class (positive, negative) is: [Input Class]

We construct a synthetic pretraining corpus through comprehensive analysis of additional time-series datasets from the UCR Time Series Classification Archive (Dau et al., 2019), specifically MoteStrain, TwoLeadECG, Wafer, PhalangesOutlinesCorrect, and Yoga. Our data curation framework includes multiple statistical approaches: trend identification via linear regression, anomaly detection using z-score analysis, and assessment of temporal stability through variability thresholds. To enhance corpus diversity, we incorporate binary (yes/no) questions targeting specific time-series characteristics. The resulting QA pairs create a rich textual representation that captures the underlying temporal and statistical properties of the data.

We use the base encoder with trained classification head as the baseline and we measure the prediction performance with accuracy.

**Graphs**  We use Graphormer (Ying et al., 2021) as the encoder model, specifically the one that was pretrained on large scale quantum chemistry graph datasets (Hu et al., 2021). Since the down-stream task (ogbg-molhiv (Hu et al., 2020)) is a molecule property prediction dataset, and with strong class imbalance (3% positive classes), we finetune the encoder on the training set to provide meaningful baselines and embeddings. We take the output prior to the classification layer of the Graphormer as the graph embedding.

We use the ogbg-molhiv (Hu et al., 2020) dataset for the graph classification task. ogbg-molhiv is a molecule property prediction dataset consisting of a total 41.12K graphs with node features and edge attributes. It has a strong class imbalance of having around 3% positive class and 97% negative class.

Hence weighted sampling is adopted in the finetuning of both the baseline Graphormer and the embedding projector to yield meaningful predictions. We use the following prompt in ICL:

(□)'s class (positive, negative) is: [Input Class]

We use the finetuned Graphormer as our baseline and use the F1 score as the performance metric due to the significant class imbalance.

### A.2  CASE STUDY: WHAT INFORMATION WAS PERCEIVED FROM PROJECTED BRAIN FMRI?

Fig. 5 illustrates the mean accuracy achieved in decoding different categories of brain activity based on fMRI data. The categories, ranging from "Physical Actions and Movements" to "Conflict, Urgency, and Change," represent diverse cognitive and perceptual domains. The grouping and corresponding questions are listed in Table 4.

The input data is noisy, and the projector is only trained with pretraining objectives, i.e., to predict the next piece of text given the current fMRI signal. We are surprised that with this pure unsupervised training, the LLM can still pick up meaningful signals from the projected embeddings. Notably, decoding tasks associated with "Physical Actions and Movements" and "Cognitive and Reflective Aspects" demonstrate higher mean accuracy, suggesting that these categories are more distinguishable based on the fMRI embeddings.

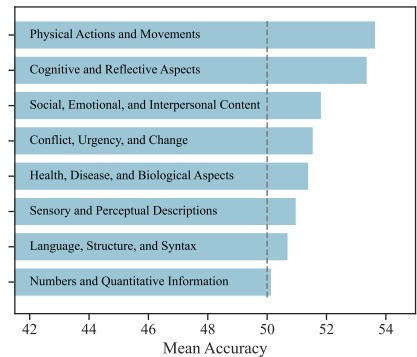

Figure 5: Analyzing LLM's understanding of projected brain fMRI embeddings after only unsupervised pretraining. We categorize the underlying questions related to the text and measure the mean accuracy for each category, highlighting the LLM's ability to interpret the embeddings, with only the next token prediction pretraining.

## A.3 TEXT CLASSIFICATION AND TEXT SUMMARIZATION CONFIG IN TABLES

Llama3.1-8B is used as the common LLM and NV-Embed-v1 is used as the text encoder for Vector-ICL. The ICL and soft-prompt methods are supplied with up to 50 shots as context. The soft-prompt tokens are trained over the entire training set.

Table 3: Hyperparameters for V-ICL training.

| Hyperparameter | Value |
|---|---|
| Learning Rate | 1e-3 |
| Learning Rate Schedule | Cosine Annealing |
| Optimizer | AdamW |
| $\beta_1$ | 0.9 |
| $\beta_2$ | 0.999 |
| Training dtype | bf16 |
| Batch Size | 128 |
| Generation Temperature | 2e-1 |

## A.4 ADDITIONAL EXPERIMENTAL RESULTS

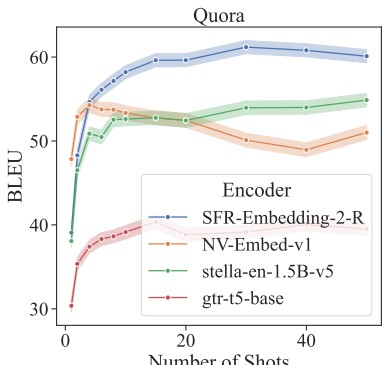
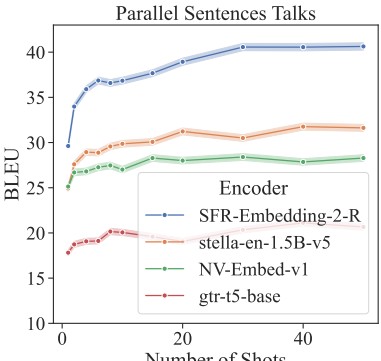

Figure 6: Text Reconstruction

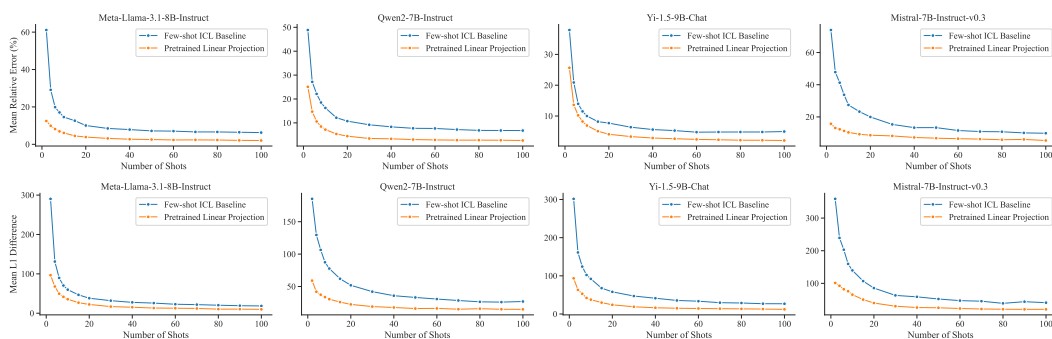

Figure 7: Function Regression - 10 digits (upper) and 3 Digits (lower)

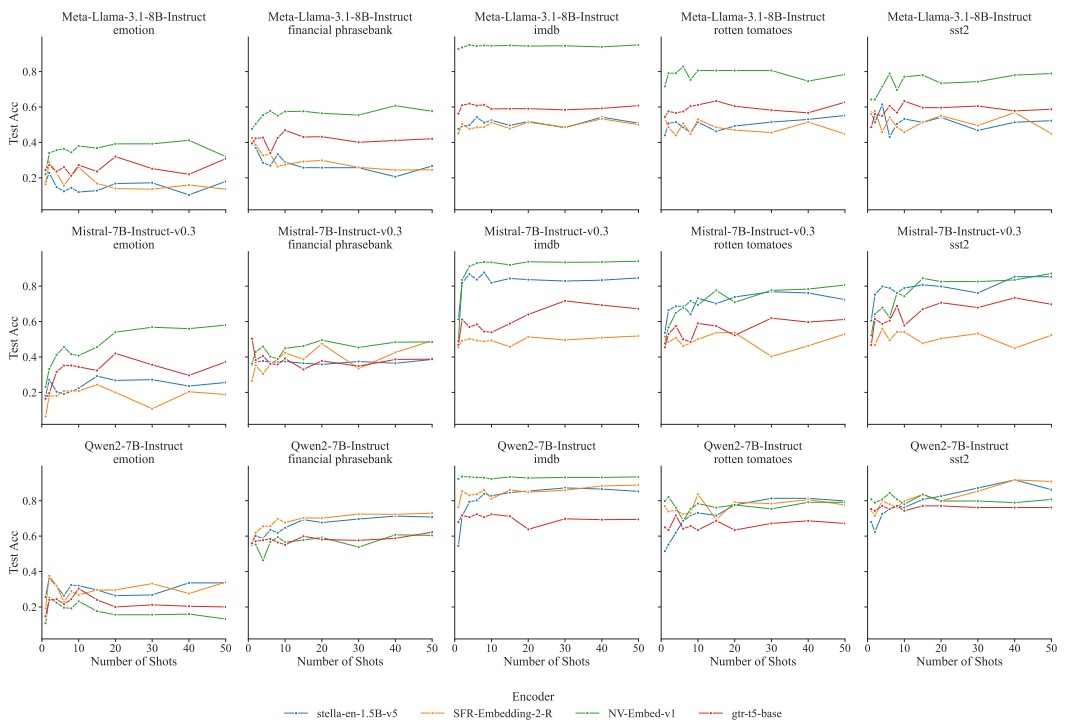

Figure 8: Text Classification - Pretrained Projectors

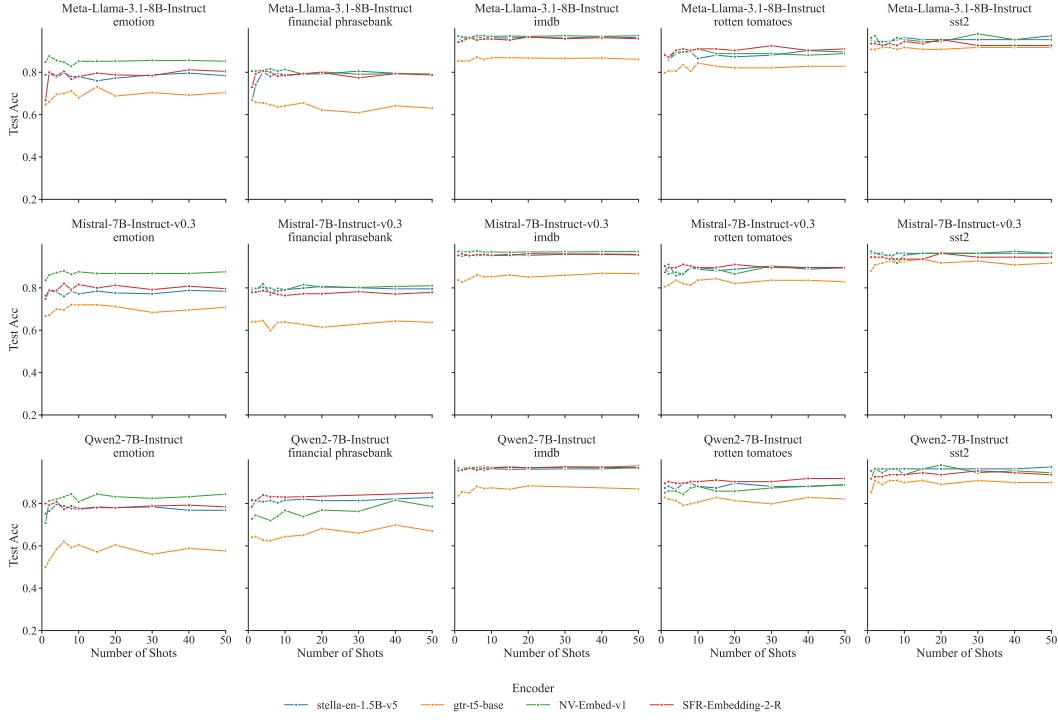

Figure 9: Text Classification - Finetuned Projectors

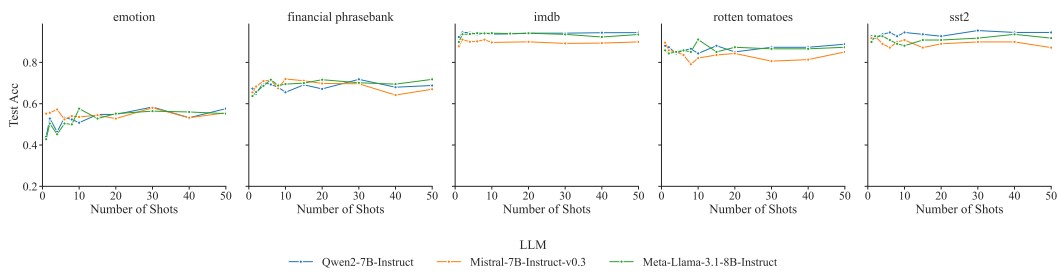

Figure 10: Text Classification - Few-shot ICL

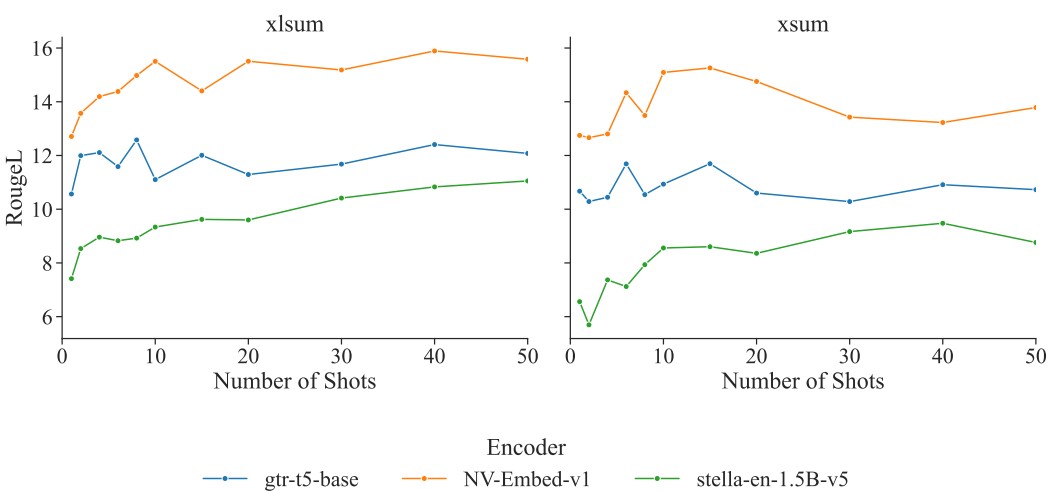

Figure 11: Text Summarization - Pretrained Projectors

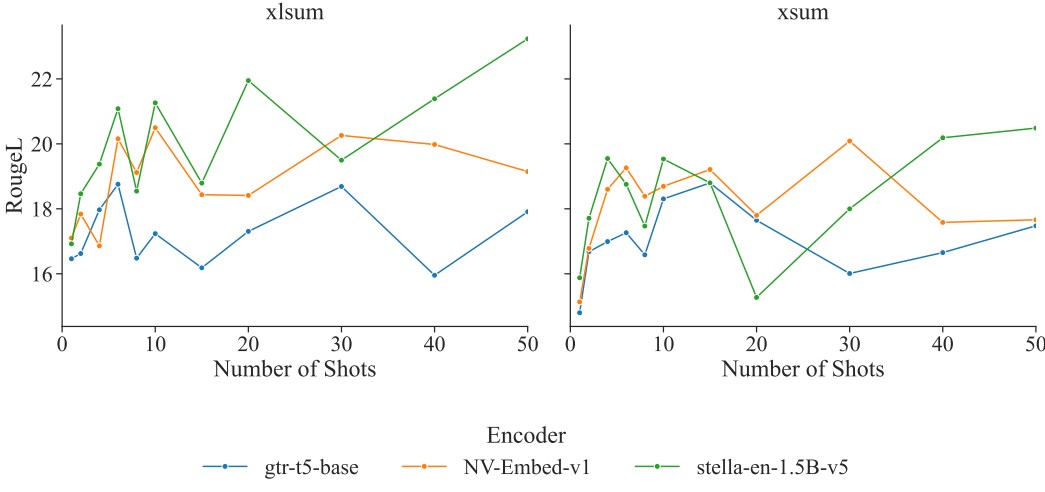

Figure 12: Text Summarization - Finetuned Projectors

## A.5 BRAIN FMRI QUESTION CATEGORIES

Table 4: Question Categories and Their Associated Questions

| Category | Questions |
|---|---|
| **Sensory and Perceptual Descriptions** | Does the input mention or describe a taste? |
| | Does the input mention or describe a sound? |
| | Does the sentence include a specific sound or auditory description? |
| | Does the input mention or describe a visual experience? |
| | Does the input mention or describe a texture? |
| | Does the sentence describe a sensory experience? |
| | Does the input mention anything related to color? |
| | Does the input mention or describe a smell? |
| | Does the input mention anything related to eyes? |
| | Does the sentence describe a visual experience or scene? |
| | Does the input describe a specific texture or sensation? |
| | Does the sentence describe a specific sensation or feeling? |
| **Social, Emotional, and Interpersonal Content** | Does the input mention anything related to arguing? |
| | Does the input mention anything related to empathy? |
| | Does the sentence involve a discussion about personal or social values? |
| | Does the input discuss a societal issue or social justice topic? |
| | Does the input mention or describe high emotional intensity? |
| | Does the sentence describe a relationship between people? |
| | Does the input mention or describe highly positive emotional valence? |
| | Does the input mention or describe highly negative emotional valence? |
| | Does the input mention anything related to conflict? |
| | Does the sentence describe a personal or social interaction that leads to a change or revelation? |
| | Does the sentence express a philosophical or existential query or observation? |
| | Does the sentence involve an expression of personal values or beliefs? |
| | Does the sentence express a sense of belonging or connection to a place or community? |
| | Is the sentence emotionally positive? |
| **Cognitive and Reflective Aspects** | Is the sentence reflective, involving self-analysis or introspection? |
| | Does the input involve planning or organizing? |
| | Does the text include a planning or decision-making process? |
| | Does the sentence convey a decision or choice made by the narrator? |
| | Does the sentence describe a personal reflection or thought? |
| | Is the input about a discovery or realization? |
| | Does the input contain a sense of ambiguity? |
| | Is the sentence providing an explanation or rationale? |
| | Does the input mention anything related to knowledge? |
| **Language, Structure, and Syntax** | Does the sentence contain a proper noun? |
| | Does the sentence include a conditional clause? |
| | Does the sentence contain a negation? |
| | Does the sentence use a unique or unusual word? |
| | Does the sentence include a direct speech quotation? |
| | Does the sentence include dialogue? |
| | Does the sentence contain a cultural reference? |
| | Does the sentence involve a recount of a social or community event? |
| | Does the input include a comparison or metaphor? |
| | Does the sentence include technical or specialized terminology? |
| | Is the sentence abstract rather than concrete? |
| | Does the sentence include an account of a miscommunication or misunderstanding? |
| | Does the text describe a mode of communication? |
| **Physical Actions and Movements** | Is the sentence conveying the narrator's physical movement or action in detail? |
| | Does the input mention anything related to motor movements? |
| | Does the sentence describe a physical action? |
| | Does the sentence describe a physical sensation? |
| | Does the sentence describe an activity related to daily life or routine? |
| | Does the input mention anything related to an action? |
| | Does the sentence involve spatial reasoning? |
| **Numbers and Quantitative Information** | Does the input mention a number less than 5? |
| | Does the input contain a number? |
| | Does the input mention a number greater than 100? |
| | Does the input mention anything related to arithmetic? |
| | Does the input mention anything related to calculation? |
| | Does the input contain a measurement? |
| **Health, Disease, and Biological Aspects** | Does the input mention anything related to diseases? |
| | Does the input mention anything related to food? |
| | Does the input mention anything related to age? |
| | Does the input mention anything related to gender? |
| | Does the input mention anything related to disgust? |
| | Does the input mention anything related to children? |
| **Conflict, Urgency, and Change** | Does the sentence involve an unexpected incident or accident? |
| | Does the input mention anything related to anger? |
| | Does the sentence convey a sense of urgency or haste? |
| | Does the sentence describe a change in a physical or emotional state? |
| | Does the sentence describe a moment of relief or resolution of tension? |
| | Does the sentence express the narrator's opinion or judgment about an event or character? |

