# OpenReview forum: "Vector-ICL: In-context Learning with Continuous Vector Representations"
_ICLR.cc/2025/Conference — ICLR 2025 Poster_

### Official Review · Reviewer_rSng · 2024-10-27

**Soundness:** 4
**Presentation:** 4
**Contribution:** 3
**Rating:** 6
**Confidence:** 4

**Summary:**

The paper explores using embeddings for examples in in-context learning (ICL), which may be more suited in non-textual tasks. The paper proposes a light-weight pretraining and finetuning scheme to obtain the encoder. The paper shows improvements over textual ICL on various non-textual tasks.

**Strengths:**

The paper identifies an interesting research problem and presents a well thought-out exploration. The idea of using embeddings for conditional generation has been explored n various indirect forms (e.g., [1]), but the paper's focus on ICL seems to be a useful contribution.

[1] Text Embeddings Reveal (Almost) As Much As Text (Morris et al., 2023)

**Weaknesses:**

1. The proposed framework is back to encoder-decoder. In this context, previous works like Flan-T5 have thoroughly explored the model's ICL capabilities [2]. While the paper differs since it focuses on a single embedding  and non-textual tasks, it's a bit odd to omit this context.

2. Intuitively vector-ICL should not work as well as textual ICL for a lot of tasks. Clearly if the task requires a lot of symbolic data ICL is strictly more powerful than the proposed approach. But the paper seems to mostly show positive results. It'd be good to do some exploration of the cases vector-ICL is worse than textual ICL and by how much.

[2] Scaling Instruction-Finetuned Language Models (Chung et al., 2022)

**Questions:**

N/A

---

> ### Author Response · Authors · 2024-11-20
>
> Thank you for your insightful feedback. We appreciate your thoughtful analysis and would like to address your points:
>
> **Encoder-Decoder Context**
>
> You raise a fair point about the connection to encoder-decoder work like Flan-T5. While our method does involve encoding and projection, there are key differences:
>
> Our focus is on enabling LLMs to process continuous vectors through lightweight projectors, rather than training a full encoder-decoder architecture. Unlike Flan-T5 which requires extensive instruction tuning, our method only trains small projectors while keeping the LLM frozen. We extend beyond text to handle arbitrary continuous vectors from various modalities
>
> We have added Flan-T5 in related work to better position our contributions in this context.
>
> **Limitations and Failure Cases**
>
> We appreciate this suggestion to provide a more balanced analysis. We agree that Vector-ICL wouldn’t be the choice for a lot of tasks, such as logic reasoning or code generation, where symbolic correctness plays an important role. But the key message of this work is that: language models can now process continuous vector context from any kind of input. We think this would open doors to lots of other interesting research and applications.

---

### Official Review · Reviewer_c8i3 · 2024-10-30

**Soundness:** 3
**Presentation:** 2
**Contribution:** 2
**Rating:** 6
**Confidence:** 4

**Summary:**

The paper is about projecting input data into an LLM embedding space, such that the LLM can perform in-context learning on the projections. The paper claims that this approach can improve the performance of the LLM on downstream tasks and opens up new application domains with Graphs, fMRI, and Time Series.

**Strengths:**

To the best of my knowledge, the proposed method is original (although some connections with soft prompt tuning should be discussed).
The paper is for the most part clear and the approach is interesting in the sense that it shows that LLMs can operate in-context on projected vectors. The results seem promising, as the learned projectors allow LLMs to tackle new tasks (graphs, fMRI etc) that are (assumed to be) not possible to tackle without projectors. The experimental part covers a wide range of tasks with different input types (8 tasks, with only 3 being classic text tasks and the proposed method beats its baselines in most cases. Clarity is generally good with some exceptions (see questions).

**Weaknesses:**

1. A weakness is that the projectors require pre-training with a language modeling objective and task-specific fine-tuning. It feels this defeats the purpose of in-context learning (i.e. not needing any training data to tackle a new task) to some extent. Authors can reflect if a change of the proposed methods name would be needed here.
2. Although the paper uses soft prompt tuning as a baseline, the relationship of the proposed approach with soft prompt tuning (Li and Liang, 2021; and follow-up work) is not explicitly discussed, i.e. as both methods learn some artificial token representations that are placed into the prompt.
3. The paper assumes that regular ICL is not applicable to Graphs, fMRI, and Time Series. While arguably not a natural fit, this assumption should be supported by some experiments, e.g. just using the numerical PCA vectors as text, or some flattened representation of the graph edges. As people have even thrown the parameters of another LM at an LM -- just in text format, I could imagine that even those non-text data types could be successfully encoded as text in a more straightforward manner without the need for training artificial tokens.
4. The experiments on time-series, graph classification and two fMRI tasks do not have a baseline, but only compare the pretrained-only projectors with fine-tuned projectors. It would be interesting to see how the proposed approach compares to some standard baseline of the respective domain, i.e. graph neural networks for graphs.


Minor point: In terms of presentation, the results description could directly link to the results, even if repetitive, e.g., Fig 3.

**Questions:**

q1: How does the proposed approach differ from soft prompt tuning (Li and Liang, 2021)?

q2: In 3.3 it is mentioned that the LLM would be trained with ``conditional generation loss''. However, in Figure 2b that is referred to, the LLM is supposed to be frozen. Could you clarify this point and explain the training process in more detail, including which components are frozen and which are updated during a) pre-training and b) fine-tuning?

q3: For Time Series (l 346), it is mentioned that you use the output of the last time step from Chronos-base. Doesn't this mean that the model is not able to use the full context of the time series but just the final state? What are the trade-offs between using the full context of the time series and just the final state?

---

> ### Author Response · Authors · 2024-11-20
>
> Thank you for your thoughtful review and constructive feedback. We appreciate your detailed analysis and would like to address each point:
>
> **We don’t necessarily require task finetuning**
>
> We appreciate this insightful observation. Our intention is to enable LLMs to process continuous context via simple projections. To clarify: Vector-ICL without supervised training outperformed few-shot ICL on 4 out of the 6 tasks when direct comparison is avaliable (numbers, molecule, brain decoding, brain classification) and corresponding baselines. We added more baselines for few-shot ICL on multi-modal inputs, and demonstrated what we expected before, that few-shot ICL is unable to directly work with such cross-modal data. And we added experiments with pretraining-only projectors on time-series with synthetic data, and it outperformed both few-shot ICL and the baseline, demonstrating the usefulness of our approach.
>
> **Relationship with Soft Prompt Tuning**
>
> Our method shares conceptual similarities with soft prompt tuning in that both involve learning embeddings that influence the LLM's behavior. However, there are key distinctions.
>
> Soft prompt tuning optimizes continuous prompts to elicit specific behaviors from LLMs on text-based tasks. Our approach learns projectors that map any form of data into the LLM's embedding space, enabling it to better handle previously challenging inputs like large numbers and molecule strings and process entirely new data modalities.
>
> We’ve added a baseline paragraph in section 4 to explain how we are using soft-prompt as a baseline for textual tasks with task-specific training.
>
> **Additional Baselines for Non-textual Inputs**
>
> We thank you for your suggestion, following this idea, we added few-shot ICL baselines by supplying the inputs in their text form (we explained how we did it in the baseline paragraph as well, in Sec. 4). We showed that the language models cannot comprehend such data as one might expect.
>
> As for directly throwing the data into LMs, we think it’s possible but not without significant limitations. For example, vision LLMs still require an encoder to encode image patches. There has been research on directly processing the pixels (PiT [1]), but any regular-sized image would impose impossible the compute and memory requirements. So we don’t think this is the significantly more promising approach compared to our encoder-based Vector-ICL.
>
> **Additional Baselines for Time-series/graphs**
>
> The current baselines are obtained via tuning the domain-specific models. We agree more domain-specific baselines would make our work better but the key message of our research is not “Vector-ICL would beat SOTA models on each domain” but rather “language models can understand continuous context via Vector-ICL”.
>
> **For your questions:**
>
> **Q1** *How does the proposed approach differ from soft prompt tuning (Li and Liang, 2021)?*
>
> We have provided our answer to this question in the last section. To reiterate, soft prompt tuning optimizes continuous prompts to elicit specific behaviors from LLMs on text-based tasks. Our approach learns projectors that map any form of data into the LLM's embedding space, enabling it to better handle previously challenging inputs like large numbers and molecule strings and process entirely new data modalities.
>
> **Q2** *In 3.3 it is mentioned that the LLM would be trained with ``conditional generation loss''. However, in Figure 2b that is referred to, the LLM is supposed to be frozen. Could you clarify this point and explain the training process in more detail, including which components are frozen and which are updated during a) pre-training and b) fine-tuning?*
>
> The encoders and LLM are always kept frozen. We thank you for pointing this out, we have made an edit to prevent confusion.
>
> **Q3** *For Time Series (l 346), it is mentioned that you use the output of the last time step from Chronos-base. Doesn't this mean that the model is not able to use the full context of the time series but just the final state? What are the trade-offs between using the full context of the time series and just the final state?*
>
> Chronos models the time-series are with T5’s encoder-decoder architecture. So we think the final state should still have all the information about the full context.
>
> [1] Nguyen, Duy-Kien, et al. "An Image is Worth More Than 16x16 Patches: Exploring Transformers on Individual Pixels." arXiv preprint arXiv:2406.09415 (2024).

---

> ### Author Response · Authors · 2024-11-23
>
> Thank you again for your thoughtful and detailed reviews of our paper. As the discussion period is approaching the end, we wanted to follow up to ensure we've fully addressed your feedback.
>
> Your comments have made the methodology and contribution clearer for our work. And if you feel we have adequately addressed your major criticisms, we would appreciate you considering updating your score accordingly.

---

> > ### Comment · Reviewer_c8i3 · 2024-11-23
> >
> > Thank you very much for the clarifications.
> >
> > Regarding the first point: I understand that Vector-ICL doesn't necessarily need task-specific fine-tuning but still the projections need to be trained in the first place for each different type of input data. Please correct me if I'm wrong – as I understood, the data to train these projections comes from the same dataset on which the approach is then evaluated (even if a different split). It leaves me wondering how the projections trained on dataset A would generalize to other datasets of the same type B,C,D without further training of the projections. I hope this clarifies my initial concern.
> >
> > Either way I believe it is an interesting approach and I agree that even if training of the projections is needed per dataset, it is lightweight compared to full fine-tuning. I will now update my ratings, yet I would appreciate a further clarification of the point above.

---

> > > ### Author Response · Authors · 2024-11-23
> > >
> > > We appreciate your continued engagement with our work and your decision to raise the rating.
> > >
> > > Your understanding is correct, when we conduct task-specific finetuning, we train the projectors on the training split of the same datasets. But in pretrained projectors, for example, the linear projectors on text, we only train the projectors on WikiText and with next-token prediction objectives. The projector-produced-vectors can directly perform various downstream ICL tasks, which is a thought-provoking result from our perspective.

---

### Official Review · Reviewer_bbMX · 2024-11-04

**Soundness:** 3
**Presentation:** 2
**Contribution:** 3
**Rating:** 6
**Confidence:** 4

**Summary:**

*Update after rebuttal*

After discussions with the authors my main concern was resolved. It appears that soft prompting with few-shot is a good baseline to demonstrate the power of Vector-ICL. I raised my score accordingly.

----------

The paper proposes a new method for in-context learning. By encoding examples into single vector representations via a separate, frozen encoder, and training a simple projection on top of the vector, LLMs are trained to perform ICL from continuous vector representations alone (Vector-ICL). The projection is trained a) via pretraining on unlabeled data and can optionally be b) finetuned on a labeled corpus.
The method is evaluated on a range of tasks, including text only tasks like text classification and summarization, as well as multi-modal tasks that utilize encoders for brain scans, graph data, or time-series.

**Strengths:**

* The proposed approach is simple to understand and implement.
* The method is evaluated on a wide range of tasks and datasets, including multiple modalities.
* Parts of the results suggests that the models perform better with more shots.
* The paper is relevant to the ICLR audience.

**Weaknesses:**

* The paper claims that with finetuning the vector-ICL method outperforms standard ICL. However, The method's baseline is standard-ICL without any finetuning, i.e. the method compares a supervised method (albeit with a weak adapter projection P) with an unsupervised one. A baseline that is obviously missing is when finetuning the base model with standard ICL, perhaps using an adapter that is equally weak. For text classification and summarization, the soft prompting baseline may be adequate, but the reason for its inclusion is never discussed.
* The method is not well-motivated. In lines 34-38 it states that many data modalities cannot be well represented in natural language, making continuous vector representations necessary for in-context learning. While this is true, many of the experiments on non-textual either don't benefit from multiple shots or need to be trained on downstream data to work (e.g. time-series classification and graph classification in Figure 3, bottom). What's the value of Vector-ICL here? If you need to finetune a model to perform a task, can it still be called in-context learning?
* The paper structure could be better. The information pertaining to a particular task is scattered all over the paper, such that I found myself scrolling back and forth a lot. For example, I did not find Figure 3 particularly helpful because the information necessary to understand it fully is located far away from it (e.g. what the horizontal bar represents). I think it would be better to focus on fewer tasks in the main body of the paper, but explain these really well. The remainder could go into the appendix.

**Questions:**

* Figure 4a: Why does the correlation vary so much between datasets and models of similar size?
* Figure 4b: What do the axes represent?
* lines 477-479: Can you expand on the block patterns and how they are explained?
* How does the finetuning work precisely? Do you finetune with multiple shots already? If so, how many?

---

> ### Author Response · Authors · 2024-11-20
>
> Thank you for your thorough review and constructive criticism of our paper. We appreciate the detailed feedback and would like to address your key concerns:
>
> **Baselines**
>
> We apologize for any confusion regarding the baselines. To clarify: for text tasks, we compare finetuned projectors against soft prompt tuning, which is indeed a supervised baseline requiring similar computational resources as our method, and pretrained projector with few-shot ICL.
> For non-text tasks, we compare finetuned projectors against tuned domain-specific encoders (like Graphformer for graphs, Chronos for time-series) as detailed in Section 4. We added few-shot ICL as baselines by transforming the inputs into text (see added Baseline paragraph in Sec. 4, we also try to make the comparison clear over there), and a pretrained projector for time-series (see details on synthetic pretraining corpus in Sec 6.3). So Vector-ICL without supervised training outperformed few-shot ICL on 4 out of the 6 tasks when direct comparison is possible (numbers, molecule, brain decoding, brain classification), which shows its potential in lots of scenarios when the input doesn’t have a natural text form.
>
> **Motivation**
>
> Thanks for your suggestions. We’ve added few-shot ICL baselines for these cross-modality tasks, and from the results we can see models can not make good use of it. We also added experiments with pretraining-only projectors on time-series, and it significantly outperformed over few-shot ICL and the baseline, demonstrating the usefulness of a simple module. And to reiterate Vector-ICL without supervised training outperformed few-shot ICL in a lot of cases. We think this provides enough motivation for this work.
>
> **Writing**
>
> We appreciate your feedback on the paper's organization. We have added a paragraph to clarify what baselines are we comparing to and reorganized the tables for better readability.
>
> **For your questions:**
>
> **Q1** *Figure 4a: Why does the correlation vary so much between datasets and models of similar size?*
>
> We think the correlation variation likely comes from:
> * Dataset characteristics: Some datasets (e.g., IMDB) contain longer texts with more complex semantic structures, making the relationship between reconstruction and downstream performance less direct
> * Model architectures: Despite similar sizes, different LLMs have varying attention patterns and token processing mechanisms that affect how well they utilize the projected embeddings
> * Encoder-LLM alignment: Some encoder-LLM pairs naturally align better in their embedding spaces (same base model, similar training data etc.), leading to more consistent performance correlations
>
> **Q2** *Figure 4b: What do the axes represent?*
>
> The axes in Figure 4b represent numbers of uniformly sampled numbers from 0 to 1e10, we’ve updated the graph with x/y axis. Thanks for bringing this up.
>
> **Q3** *lines 477-479: Can you expand on the block patterns and how they are explained?*
>
> The block patterns in the distance matrix emerge from our number representation scheme and how it interacts with the projector's learning. Each number is represented as a concatenated one-hot encoding of its digits - for a 10-digit number, this creates a 10×10 matrix (one-hot vector for each digit), flattened into a 100-dimensional vector. The projector learns to preserve these numerical relationships during training, maintaining the hierarchical structure of the decimal system. This organization helps the LLM process numerical relationships more effectively, as the distance patterns reflect natural place-value relationships between numbers. We’ve added additional explanations in the analysis section as well.
>
> **Q4** *How does the finetuning work precisely? Do you finetune with multiple shots already? If so, how many?*
>
> The process of finetuning can be found in Sec 3.3. It is as simple as training for any conditional generation data, where we freeze encoder and LLM, and train the projector with task-specific datasets with the conditional generation loss.
> Yes, we generally use 128-shot in finetuning,

---

> > ### Comment · Reviewer_bbMX · 2024-11-22
> >
> > Thank you for the clarifications.
> >
> > Regarding your choice of soft prompting as baseline (and possible also in regards to Q4): Does this baseline still contain the same few-shot prompt as Vector-ICL and standard ICL, namely a couple of shots? This would be needed for having a somewhat fair comparison between vector-shots and textual shots, IMO.

---

> > > ### Comment · Reviewer_bbMX · 2024-11-25
> > >
> > > Then the soft prompting baseline seems to be what I was looking for, resolving my main concern with the paper. I raised my score accordingly.

---

> > > > ### Author Response · Authors · 2024-11-25
> > > >
> > > > Thank you for the update and the engaging discussion! We appreciate your thorough feedback, which helped us improve the quality of our work.

---

> ### Author Response · Authors · 2024-11-22
>
> Thank you for your follow-up.
>
> Yes, we controlled the soft-prompt tuning to have the same contextual examples as Vector-ICL/few-shot ICL. We've updated Appendix A.3 to clarify this experimental setup.
>
> We appreciate your thorough review and we look forward to addressing any remaining questions or concerns you might have.

---

> ### Author Response · Authors · 2024-11-23
>
> Thank you again for taking the time to review our paper and provide thoughtful feedback! As the discussion period is coming to an end, we wanted to check if there are any remaining concerns or clarifications we can address to further improve our work.
>
> Your comments have helped us significantly improve our work, particularly in improving how we motivate our work and clarifying how we compare against the baselines. If we’ve already addressed your major concerns, we kindly hope you’ll consider updating your rating to reflect this.

---

### Official Review · Reviewer_5yn3 · 2024-11-05

**Soundness:** 3
**Presentation:** 3
**Contribution:** 3
**Rating:** 6
**Confidence:** 3

**Summary:**

The paper studies the feasibility of vector-ICL, that extends the in-context learning capabilities of LLMs to continuous vectors. Authors use light-weight projectors to align embedding space of input text embedding with LLM. To train the projectors, they use general language modeling objective followed by task-specific objectives. Authors experimented with multiple tasks and modalities, e.g., text classification, summarization, time-series classification, fMRI decoding, etc.

**Strengths:**

- The concept of using vectors for in-context learning is a new exploration, and the proposed methods for vector-ICL and evaluations show  promising results.

- Using light-weight trainable projectors with simple pre-training on general task is also not very expensive and can be integrated as a part of general LLM pre-training.

- Experiments cover a wide range of tasks and modalities.

**Weaknesses:**

- Method lacks depth. Specifically, replacing any length text for any complexity task with a single embedding may not be sufficient. Including an ablation where text is replaced with a series of vectors (one vector per sentence/ chunk) would be helpful.

- Currently, the method requires task-specific fine-tuning to outperform token-ICL. I think authors should explore RLHF/ instruction finetuning datasets/ objectives to avoid task-specific finetuning.

**Questions:**

- How do you see vector-ICL applied in practical scenarios, considering decreasing inference cost and increasing context length of LLMs?  Given your experiments show that without task-specific fine-tuning, vector-ICL consistently outperforms tokens-based ICL?

- Did you explore instruction tuning after pre-training? Authors should discuss the tradeoffs between task-specific finetuning and more general approaches like instruction tuning/ human preference training.

---

> ### Author Response · Authors · 2024-11-20
>
> Thank you for your careful review and insightful feedback on our paper. We greatly appreciate your recognition of the novelty and promise of our approach, as well as your constructive suggestions for improvement. We would like to address your key points:
>
> **Single-Vector v.s. Multi-Vector**
>
> We agree that the single vector approach may not be optimal, and we have included it as our future direction in the discussion section (*“Also, we only experimented with single-token encoders, while there exist encoders that produce variable-sized embeddings, that can potentially be more powerful and flexible.”*). Overlooking from the RAG lines of work, chunking the input, and encoding them as multiple vectors is definitely possible but non-trivial. We want this paper to be a starting point for exploiting continuous vector context, which opens up research questions like this.
>
> **Instruction Tuning & Baselines**
>
> We thank you for the suggestion. Firstly, we created such a synthetic instruction dataset with time-series data, where natural text pairs are difficult to obtain. We utilized the statistical properties of the time series such as its stability and overall trend (more details in Sec 6.3). We then use this dataset to pretrain the projector for time-series, turns out the pretrained projector + LLM can generalize to the classification task (see the new Figure 4).
>
> Secondly, we added more baselines for few-shot ICL on multi-modal inputs, and demonstrated what we expected before, that few-shot ICL is unable to directly work with such data. So to clarify, Vector-ICL with pretraining is more performant than few-shot ICL on 4 out of the 6 tasks when direct comparison is possible (numbers, molecule, brain decoding, brain classification), which shows its potential in lots of scenarios when the input doesn’t have a natural text form.
>
> **For your questions:**
>
> **Q1** *How do you see vector-ICL applied in practical scenarios, considering decreasing inference cost and increasing context length of LLMs? Given your experiments show that without task-specific fine-tuning, vector-ICL consistently outperforms tokens-based ICL?*
>
> We have clarified the performance issue. And we want to highlight that our work is not about efficiency per se (compressing long inputs into single tokens), it is about how any form of data can be utilized by LLMs via encoding and projection. As we have shown in the baselines, supplying time-series/graphs/brain fMRI naively into the language models does not work very well. So we are confident that Vector-ICL can be of practical use in many such scenarios.
>
> **Q2** *Did you explore instruction tuning after pre-training? Authors should discuss the tradeoffs between task-specific finetuning and more general approaches like instruction tuning/ human preference training.*
>
> Thank you for this suggestion. We have explored instruction tuning through a synthetic instruction corpus for time-series data, where natural text-data pairs are difficult to obtain. Our pipeline generates instruction-like data by analyzing statistical properties (trend, anomalies, stability) of time-series and converting them into natural language QA pairs. Results show that projectors pretrained on this synthetic corpus already outperform both few-shot ICL and domain-specific baselines. We’ll leave the text-based instruction tuning / RLHF for the future, as there’s lots of variability in such studies.

---

> > ### Comment · Reviewer_5yn3 · 2024-11-22
> > **clarifications**
> >
> > Thank you for clarifications,
> >
> > - about how any form of data can be utilized by LLMs via encoding and projection.
> > IMO this is well proven by VLMs, as well as some works on Graphs + LLM works like GraphLlava
> >
> > - By instruction tuning, I meant standard datasets such as Supernatural instruction, Alpaca, dolly etc.

---

> > > ### Author Response · Authors · 2024-11-22
> > >
> > > Thank you for your response!
> > >
> > > Regarding your first point, our work is inspired by previous research, such as LLAVA or the ones you mentioned. But different from them, we are the first to explore the following: without tuning the encoder or language model, LLMs can process and utilize those projected embeddings (often a linear transformation would work) as contextual demonstrations, for a wide range of modalities and tasks, which we think is a pretty exciting finding.
> > >
> > > We thank you for your clarification, we've added pretraining with instruction tuning as a potential future work in Section 7 to reflect that.
> > >
> > > We hope this addresses your questions. We welcome any further discussion or clarification you may need.

---

> > > > ### Comment · Reviewer_5yn3 · 2024-11-28
> > > >
> > > > Thank you for the further clarifications.

---

### Meta-Review · Area_Chair_TxLK · 2024-12-17

**Metareview:**

This paper proposes Vector-ICL which encodes input demonstrations into continuous vectors via pretrained projectors and fine-tuning process to elicit LLMs' capability of understanding both vector and textual inputs. This strategy enables effective use of the ICL framework in domains lacking textual descriptions such as time-series, molecules, graphs. Extensive experiments on various domains and tasks (including across modalities) demonstrate the advantage of Vector-ICL over traditional few-shot ICL.

Strengths:
- Extending ICL capability to the continuous space is interesting and novel to some extent.
- Vector-ICL could obviously benefit diverse application domains where natural texts are missing or not feasible, such as molecules, time series data, etc.
- Extensive experiments have been conducted on a number of tasks and application domains, demonstrating a clear advantage of the framework over few-shot textual ICL. It is also shown that projector pretraining could have some extent of generalization beyond the training domain.

Weaknesses:
- This approach still requires a certain level of fine-tuning over the task-specific training data, limiting its generalizability and scalability towards general usecases (although the authors have shown that the method could outperform few-shot ICL in 4 out of 6 tasks without fine-tuning). Further exploration on more generic pretraining and evaluation without fine-tuning could be more interesting.
- Using a single vector to encode an input sample lacks capacity in representing more complex semantic meanings which could limit the adoption of Vector-ICL in complex scenarios or application domains. But given that this work pioneers the exploration of combining continuous vectors with ICL, it is acceptable to leave it as a future work.
- More discussions and comparisons between Vector-ICL and soft prompt tuning should be added, given that both paradigms combine continuous vectors with language generation.

**Additional Comments On Reviewer Discussion:**

- Many reviewers raise the issue that Vector-ICL still requires task-specific fine-tuning, which should not be the purpose of ICL when it was originally proposed. As a response, the authors presented the performance comparison between only using pretrained projector and few-shot ICL and showed the advantage of Vector-ICL on 4 out of 6 tasks. This is very helpful in demonstrating the usability of this method without task-specific finetuning.
- Reviewer also raises the question of instruction tuning under this setting. The authors added additional experiments showing how pretraining can generalize to time-series classification tasks, which partially addresses the reviewer's concern.
- Another issue is lack of comparison with baselines such as soft prompt tuning and feeding symbolic input into ICL. As a response, the authors further added experiments using these missing baselines and demonstrated a consistent performance advantage of the proposed method.

Overall, the authors have addressed most of the issues raised by the reviewers, as evidenced by the increase of the evaluation score. I can see the validity of these questions and how these questions are addressed and enhance the quality of this work.

---

### Decision · Program_Chairs · 2025-01-22

Accept (Poster)